# The burden of postpartum depression and its socio-demographic and obstetric correlates among parturient in Bangladesh: A cross-sectional study

Md Foyjul Islam[1]*, Sirajam Munira[1,2], Syeda Tasnuva Maria[1], Quazi Ahmed Zaki[1], Tahmina Shirin[1]

**1** Institute of Epidemiology Disease Control and Research (IEDCR), Dhaka, Bangladesh, **2** National Gastroliver Institute and Hospital, Dhaka, Bangladesh

* drislam0666@gmail.com

## Abstract

Postpartum depression (PPD) affects 10–15% of women globally and up to 35% in Bangladesh, yet remains under-researched across different healthcare settings. This cross-sectional study aimed to assessed the prevalence and its socio-demographic and obstetric correlates among 540 women attending tertiary, secondary, and primary healthcare centers for routine post-natal care in Bangladesh, 4–12 weeks after delivery. The Edinburgh Postnatal Depression Scale (EPDS) was used to screen for PPD (cut-off ≥10), and socio-demographic, reproductive, and obstetric data were collected through structured questionnaires. Logistic regression identified significant predictors of PPD. The prevalence of PPD was 47.78% (95% CI: 43.49–52.09), with 29.07% experiencing major depression (EPDS ≥13). Women engaged in labor work (AOR = 5.17, 95% CI: 1.70–15.70, p = 0.01), having a previous history of depression (AOR = 3.38, 95% CI: 2.17–5.28, p < 0.01), irregular menstruation (AOR = 3.58, 95% CI: 1.39–9.18, p = 0.01), a history of abortion (AOR = 1.73, 95% CI: 1.03–2.93, p = 0.03), and complications during pregnancy (AOR = 2.96, 95% CI: 1.95–4.50, p < 0.01) were at significantly higher risk of PPD. Furthermore, perceiving pregnancy as average (AOR = 2.09, 95% CI: 1.16–3.76, p = 0.01) or difficult (AOR = 3.47, 95% CI: 1.85–6.49, p < 0.01), and delivery as average (AOR = 2.51, 95% CI: 1.41–4.47, p < 0.01) or difficult (AOR = 1.77, 95% CI: 1.04–3.02, p = 0.03), were also associated with increased risk. These findings highlight that PPD is highly prevalent among Bangladeshi women, with multiple socio-demographic and obstetric risk factors. Integrating EPDS screening into routine postnatal care could enable early detection and timely intervention, thereby improving maternal mental health outcomes.

**Data availability statement:** The datasets analyzed in the current study are not publicly available due to data privacy restrictions but are available upon reasonable request from the Institute of Epidemiology, Disease Control and Research (IEDCR), Dhaka, Bangladesh (email: iedcrit@gmail.com).

**Funding:** This work was supported by the mini-grant award from the "Supporting Operational and Implementation Research Addressing the Burden of Non-Communicable Diseases in the Eastern Mediterranean Region – 2023–2024" program, provided by Global Health Development | Eastern Mediterranean Public Health Network (GHD|EMPHNET) in partnership with the Training Programs in Epidemiology and Public Health Interventions Network (TEPHINET) (Grant awarded to MFI). The funders had no role in study design, data collection and analysis, decision to publish, or preparation of the manuscript. The findings and conclusions presented in this report are those of the authors and do not necessarily reflect the views of EMPHNET or TEPHINET.

**Competing interests:** The authors have declared that no competing interests exist.

## Introduction

Post-partum depression (PPD) is the most common psychological condition after childbirth that leaves considerable harmful effects on the physical and social well-being of mothers, newborn infants, families as well as community. This condition develops in one of the most critical periods of a woman's life when the mother is going through a new adaptation with hormonal and anatomical changes [1,2]. Postpartum depression (PPD) can develop anytime within the first year after delivery and may persist for several years; if left untreated, it can increase the risk of severe mental health conditions, including postpartum psychosis, suicide, and, in rare cases, infanticide [3–5]. Usually, PPD develops within the first six weeks after delivery and manifests as sadness, low mood, severe mood swings, hopelessness, poor bonding with baby, changes in appetite or sleep pattern, feelings of inadequacy or guilt as a parent, and thoughts of harming oneself or the baby [6].

The estimated global prevalence of PPD is 10–15%, but considering the geographical location, race, ethnicity, country or region development, and country or region income, a great variation is observed (0.5-60.8%) [7,8]. The prevalence seems to be lower in higher-income countries than in lower-income countries. A systematic review revealed that Southern Africa has the highest PPD prevalence (39.96%), followed by Southern Asia (22.32%), South America (21.71%), Western Asia (19.83%), Northern Africa (18.7%), Eastern Asia (17.39%), Northern America (17.01%), and South-Eastern Asia (13.53%)) [9]. In Asian countries, prevalence estimates range from 3.5% to 63.3% [10], with studies in India reporting rates between 11–16% within 14 weeks postpartum [11]. Similarly, in rural Bangladesh, multiple small studies have indicated a prevalence rate of 18–35% [12].

Despite the high prevalence of PPD in low-income countries like Bangladesh, research on PPD has gained minimal attention. More importantly, PPD is preventable and treatable with less risky non-pharmacological methods if the risk for PPD is detected early [13]. As a densely populated country with resource constraints, there is a scarcity of multi-tier, facility-based data on PPD prevalence and its correlates. Integrating routine PPD screening into national postnatal care (PNC) guidelines offers a feasible approach to addressing postpartum mental health. The Edinburgh Postnatal Depression Scale (EPDS) is a globally validated tool with high sensitivity and specificity, quick administration (approximately five minutes), and simple scoring. It is effective in detecting depression severity and serves as a secondary prevention tool by facilitating early intervention [14–16] While high-income countries incorporate perinatal depression screening into routine postnatal care, Bangladesh faces significant barriers, including limited awareness among healthcare providers and the absence of a standardized screening approach. Many women lack knowledge about PPD, making it difficult to distinguish between postpartum blues and clinical depression. Consequently, affected mothers often endure recurrent episodes of depression, disrupted family life, and long-term psychological distress [17,18]. Even when diagnosed, stigma, and misconceptions about mental health such as consulting traditional healers (kabiraj, ojha) or local village doctors can delay access to seeking care [19–22].

The findings from this study provide baseline evidence that could inform the future integration of PPD screening into routine postnatal care (PNC) services. Incorporating validated tools such as the EPDS—widely recognized as feasible, cost-effective, and practical into PNC visits may enable early detection, timely intervention, and improved maternal mental health outcomes [23,24].Regular PPD assessments could help quantify the true burden of the condition, identify high-risk mothers, and guide the development of efficient referral systems for maternal mental health support.

Therefore, the aim of this study was to determine the prevalence of postpartum depression (PPD) and its socio-demographic and obstetric correlates among women receiving care at three tiers of health facilities in Bangladesh, generating baseline evidence to inform potential future integration of PPD screening into routine postnatal care.

## Materials and methods

### Study design and setting

This cross-sectional study was conducted to assess postpartum depression (PPD) among women attending postnatal care (PNC) within 4 to 12 weeks after delivery. The study was carried out at three levels of healthcare facilities in Dhaka and Manikganj, Bangladesh, ensuring representation from different tiers of the healthcare system. At the tertiary level, the study was conducted at Mugda Medical College Hospital, Dhaka, Maternal and Child Health Training Institute, Azimpur, Dhaka represented Secondary level hospital, while the Upazila Health Complex in Singair, Manikganj represented subdistrict-level study was. The study participant recruitment was conducted over from 15 September 2024 to 15 November 2024.

### Study population and eligibility criteria

The study population comprised postpartum women who received PNC services at the selected healthcare facilities. Inclusion criteria included women within 4 to 12 weeks postpartum with live baby, and those willing to participate. Women with pre-existing psychiatric disorders, chronic diseases, intrauterine or infant death, newborns with severe illnesses, or those unable to provide informed consent were excluded from the study. These women were excluded because acute bereavement or distress associated with severe neonatal illness could influence EPDS scores independently of postpartum depression, potentially confounding results.

### Sample size and sampling method

A total of 540 participants were selected, with 180 from each healthcare tier. The sample size was determined using Epi Info version 7, assuming a PPD prevalence of 33% with a 95% confidence interval, and a 5% margin of error, yielding an initial estimated sample size of 340. Adjustments for a 5% non-response rate and a design effect of 1.5 resulted in a final sample size of 540. We selected a 5% non-response rate based on prior national maternal health surveys in Bangladesh, which have demonstrated high response rates for face-to-face facility-based interviews [25].A consecutive sampling method was employed, selecting every eligible postpartum woman attending the PNC corners at the selected facilities within the data collection period. Equal numbers were sampled from each tier to ensure balanced representation and comparability across facility levels.

### Data collection

Data were collected using a semi-structured questionnaire and the validated Edinburgh Postnatal Depression Scale (EPDS), which was translated into Bengali for ease of administration. It was translated into Bangla following a standardized forward–backward translation process. Two bilingual experts independently translated the English version into Bangla, and another set of translators, blinded to the original version, back-translated it into English. Discrepancies were reviewed and resolved by an expert panel comprising psychiatrists, public health specialists, and linguists to ensure

semantic and conceptual equivalence. The pre-final version was pilot-tested among postpartum women to assess clarity, cultural appropriateness, and comprehension, leading to minor refinements before use in the study. The questionnaire captured sociodemographic information, pregnancy and delivery history, and mental health factors. The EPDS is a 10-item self-report screening tool with a total score range of 0–30, widely used to identify postpartum depressive symptoms. In Bangladesh, the Bangla version has been validated against SCID diagnoses, demonstrating high sensitivity (89%) and specificity (87%) at a cutoff of ≥10, with good internal consistency (Cronbach's α = 0.84) [26,27]. Participants with an EPDS score greater than 10 were classified as having PPD. The severity of PPD was further categorized into mild (EPDS score 10–12) and major (EPDS score ≥13). Women scoring ≥10 on EPDS were advised to seek mental health support. Two weeks after screening, follow-up phone calls were made to assess adherence to recommendations.

To ensure high-quality data collection, a two-day training session was conducted for nurses and PNC staff at each healthcare facility. Training covered research objectives, questionnaire administration, and the EPDS scoring process. Hands-on practice and role-playing were used to enhance understanding. Three trained personnel at each site conducted data collection over two months, with periodic supervision by investigators.

Anthropometric measurements were obtained using standardized procedures and calibrated equipment. Participants' weight was measured using a calibrated digital scale while standing barefoot, ensuring even weight distribution. Height was recorded using a stadiometer with participants standing upright against a vertical backboard. Regular calibration of equipment and adherence to standardized protocols ensured accuracy. We reported that the expected dropout rate was 5%; in practice, 27 participants (5% of the sample) were included in the pilot pre-test, and no additional participants withdrew during the main study. Missing data were minimal (<1%) and handled using complete case analysis.

## Statistical analysis

After data collection, all responses were reviewed for completeness and consistency before being coded and entered into Microsoft Excel. Data analysis was performed using StataIC 17. Descriptive statistics were used to summarize sociodemographic and reproductive history variables. Univariate and multivariate logistic regression analyses were conducted to assess associations with PPD, with results reported as odds ratios and 95% confidence intervals. A p-value of <0.05 was considered statistically significant.

## Quality assurance

The questionnaire was pre-tested on 5% of the sample (n = 27) to evaluate feasibility, clarity, and reliability. Inter-rater reliability was assessed in this subsample during the pilot phase, with two trained research assistants independently administering the Bangla EPDS, blinded to each other's ratings. Agreement for PPD caseness (EPDS ≥ 13) was assessed using Cohen's κ (κ = 0.86), and agreement for total scores using a two-way random-effects ICC (2,1), both with 95% CIs. Necessary modifications were made based on pre-test feedback. Throughout data collection, daily supervision ensured adherence to protocols. Data quality control measures included periodic audits and consistency check by investigators. Standardized procedures were followed for anthropometric measurements, and training sessions reinforced data collection techniques to minimize interviewer bias.

## Ethical considerations

Ethical approval was obtained from the Institutional Review Board (IEDCR/IRB/2024/05). Official permissions were secured from the directors of the selected medical institutions. Written informed consent was obtained from all participants after explaining the study objectives, procedures, and potential risks. Participants were assured of confidentiality, voluntary participation, and the right to withdraw at any time without consequences. Data were securely stored on a password-protected laptop accessible only to authorized investigators. All study-related documents will be retained for five

years in compliance with Ministry of Health and Family Welfare policies, ensuring continued confidentiality and integrity of the research data.

## Results

This study included 540 postpartum women receiving postnatal care at tertiary, secondary, and primary healthcare facilities.

### Sociodemographic characteristics of study participants

The median age was 25 years (IQR: 21–28), with a higher median age among those with postpartum depression (PPD) (26 years; IQR: 22–30) compared to those without PPD (24 years; IQR: 20–27). Most participants (96.67%) were Muslim, and 52.78% had secondary education. A higher proportion of women without PPD (91.86%) were housewives compared to those with PPD (85. 66%).The majority (79.81%) lived with their husbands, while 13.70% had husbands residing abroad. Nuclear (50.74%) and joint families (49.26%) were nearly equally represented. Husbands' education levels varied, with 42.41% completing secondary education and 20.37% attaining graduation or higher. Business (35.74%) and private service (24.07%) were the most common occupations. Socioeconomic status revealed that 43.15% belonged to lower class, 46.48% in the lower middle class, and 10.37% in the upper middle class. A higher proportion of PPD cases (64.34%) were from rural areas. History of previous depression was more prevalent among women with PPD (43.02% vs. 18.44%), and a family history of mental illness was slightly higher in the PPD group (10.47% vs. 7.09%). BMI distribution showed 57.78% with normal weight, 23.52% overweight, 10.19% obese, and 8.52% underweight, with no major differences between groups. [Table 1]

### Reproductive Pregnancy and delivery related characteristics of study participants

Among the participants, multiparity was more common in the PPD group (68.99%) compared to the non-PPD group (57.45%). Menstruation characteristics were similar between groups though irregular menstruation was more frequent in the PPD group (9.30% vs. 3.55%). Early marriage (<18 years) was prevalent in both groups (65.12% vs. 65.25%), as was first delivery before 20 years (59.30% vs. 64. 18%).Participants with PPD had more children (≥2 children: 69.99% vs. 57.45%) and a slightly shorter birth interval (≤24 months: 32.02% vs. 35.80%). Previous history of abortion was significantly higher in the PPD group (26.36% vs. 14.18%), with more induced abortions (41.18% vs. 37.50%) and more D&C procedures (11.76% vs. 2.50%). Unplanned pregnancies were more common in the PPD group (66.67% vs. 56.38%). Gestational age at birth was similar between groups, though twin pregnancies were slightly less frequent among those with PPD (2.71% vs. 4.61%). The gender distribution of neonates was generally balanced; however, a slightly higher number of female neonates were born to mothers experiencing PPD (52.71% vs. 46. 81%).Family support was lower in the PPD group (74.42% vs. 82.27%), and fewer women perceived pregnancy as "easy" (13.57% vs. 31.91%). [Table 2]

### Prevalence and distribution of PPD among study participants

Among the 540 participants, 47.78% (95% CI: 43.49–52.09%) screened positive for postpartum depression (PPD), defined as EPDS >10. [Fig 1A] The severity of PPD among the participants showed 18.70% experienced mild PPD (EPDS score 10–12), while a significant 29.07% suffered from major PPD (EPDS score ≥13). As per EPDS scoring, 18.70% women experienced mild PPD (EPDS score 10–12), while 29.07% suffered from major PPD (EPDS score ≥13). [Fig 1B]

The prevalence of postpartum depression (PPD) varied across healthcare facilities, with 54.14% at the Medical College Hospital, 41.90% at the Maternal and Child Hospital, and 47.22% at the Upazila Health Complex. However, the Chi-square test ($X^2 = 5.44$, p = 0.06) reveals no statistically significant differences in PPD prevalence between the sites. [S1 Table]

**Table 1. Characteristics of study participants (N = 540).**

| Variable | Overall N = 540 N (%) | No PPD N = 282 N(%) | PPD N = 258 N(%) |
|---|---|---|---|
| **Age Group** | | | |
| ≤24 | 256 (47.4) | 101(44.29) | 155(54.96) |
| 15-29 | 175 (32.4) | 87(30.85) | 88(34.11) |
| ≥30 | 109 (20.2) | 40(14.18) | 69(26.74) |
| **Median (IQR)** | 25 (21-28) | 24 (20-27) | 26 (22-30) |
| **Religion** | | | |
| Islam | 522 (96.67%) | 272(96.45) | 250(96.90) |
| Hinduism | 18(3.33) | 10(3.55) | 8 (3.10) |
| **Educational Level** | | | |
| No formal education | 29(5.37) | 16(5.67) | 13(5.04) |
| Primary | 61(11.30) | 35(12.41) | 26(10.08) |
| Secondary | 285(52.78) | 147(52.13) | 138(53.49) |
| Higher Secondary | 81(15.00) | 43(15.25) | 38(14.73) |
| Graduate and above | 84(15.56) | 41(14.54) | 43(16.67) |
| **Occupation** | | | |
| Housewife | 480(88.89) | 259(91.86) | 221(85.66) |
| Govt. Service | 22(4.07) | 11(3.90) | 11(4.26) |
| Private Service | 15(2.78) | 6(2.13) | 9(3.49) |
| Labor | 22(4.07) | 6(2.13) | 13(6.20) |
| Business | 1(0.19) | 0(0.00) | 1(0.39) |
| **Husband living Status** | | | |
| Living Together | 431(79.81) | 228(80.85) | 203(78.68) |
| Dead | 9(1.67) | 5(1.77) | 4(1.55) |
| Living in different place | 26(4.81) | 16(5.67) | 10(3.88) |
| Living Abroad | 74(13.70) | 33(11.70) | 41(15.89) |
| **Family Type** | | | |
| Nuclear | 274(50.74) | 152(53.90) | 122(47.29) |
| Joint | 266(49.26) | 130(46.10) | 136(52.71) |
| **Husband Educational Level** | | | |
| No formal education | 41(7.59) | 24(8.51) | 17(6.59) |
| Primary | 72(13.33) | 38(13.48) | 34(13.18) |
| Secondary | 229(42.41) | 118(41.84) | 111(43.02) |
| Higher Secondary | 88(16.30) | 44(15.60) | 44(17.05) |
| Graduate and above | 110(20.37) | 58(20.57) | 52(20.16) |
| **Husband Occupation** | | | |
| Unemployment | 26(4.81) | 12(4.26) | 14(5.43) |
| Govt. Service | 67(12.41) | 37(13.12) | 30(11.63) |
| Private Service | 130(24.07) | 78(27.66) | 52(20.16) |
| Labor | 124(22.96) | 57(20.71) | 67(25.97) |
| Business | 193(35.74) | 98(34.75) | 95(36.82) |
| **Economic Status (Monthly income)** | | | |
| Lower class | 233(43.15) | 130(46.10) | 103(39.92) |
| Lower Middle Class | 251(46.48) | 129(45.74) | 122(47.29) |
| Upper Middle Class | 56(10.37) | 23(8.16) | 33(12.79) |

*(Continued)*

 

**Table 1.** (Continued)

| Variable | Overall N = 540 N (%) | No PPD N = 282 N(%) | PPD N = 258 N(%) |
|---|---|---|---|
| **Residence** | | | |
| Rural | 198(36.67) | 176(62.41) | 166(64.34) |
| Urban | 342(63.33) | 106(37.59) | 92(35.66) |
| **History of Previous Depression** | | | |
| Present | 163(30.19) | 52(18.44) | 111(43.02) |
| Absent | 377(69.81) | 230(81.56) | 147(56.98) |
| **History of Mental Illness in Family** | | | |
| Yes | 47(8.70) | 20(7.09) | 27(10.47) |
| No | 473(87.59) | 251(89.01) | 222(86.05) |
| Don't Know | 20(3.70) | 11(3.90) | 9(3.49) |
| **BMI** | | | |
| Underweight (< 18.5 kg/m$^2$) | 46(8.52) | 25(8.87) | 21(8.14) |
| Normal (< 18.5 kg/m$^2$ to 24.99 kg/m$^2$) | 312(57.78) | 159(56.38) | 153(59.30) |
| Overweight (24.99 to 29.99.kg/m$^2$) | 127(23.52) | 71(25.18) | 56(21.71) |
| Obese (>29.99 kg/m$^2$) | 55(10.19) | 27(9.57) | 28(10.85) |

Regarding depression severity based on EPDS categories, 36.46% of participants at the Medical College Hospital had major depression, compared to 22.91% at the Maternal and Child Hospital and 27.78% at the Upazila Health Complex. The percentage of those with possible depression ranged from 17.68% to 19.44% across the sites. The Chi-square test ($X^2 = 5.44$, p = 0.06) again shows that the differences in depression severity across the sites are not statistically significant. **[S2 Table]**

## Association of PPD with various characteristics

The univariate and multivariate logistic regression analyses identified several significant sociodemographic, reproductive, and pregnancy-related factors associated with postpartum depression (PPD), This study found women aged ≥30 years had significantly higher odds of developing PPD compared to those aged ≤24 years (COR = 2.65, 95% CI: 1.67 - 4.21, p < 0.01; AOR = 2.42, 95% CI: 1.43 - 4.10, p < 0.01). Women from the upper middle class having increased odds of PPD compared to those from the lower class (COR = 1.81, 95% CI: 1.00 - 3.27, p = 0.04; AOR = 1.84, 95% CI: 1.02 - 3.31, p = 0.04). Women engaged in labor-intensive occupations had a significantly higher likelihood of PPD compared to housewives (COR = 3.12, 95% CI: 1.20 - 8.12, p = 0.01; AOR = 5.17, 95% CI: 1.70 - 15.70, p = 0.01). A previous history of depression emerged as one of the strongest predictors of PPD (COR = 3.34, 95% CI: 2.26 - 4.93, p < 0.01; AOR = 3.38, 95% CI: 2.17 – 5.28, p < 0.01). **[Tables 3 & Table 5]**

Multiparous women had higher odds of developing PPD compared to primiparous women (COR = 1.65, 95% CI: 1.16 - 2.35, p = 0.01; AOR = 1.79, 95% CI: 1.22 - 2.63, p = 0.003). Women with more than three children were at a significantly greater risk of developing PPD (COR = 2.50, 95% CI: 1.04 - 5.99, p = 0.04; AOR = 2.68, 95% CI: 1.10 - 6.56, p = 0.03). Those with a history of always irregular menstruation before pregnancy had significantly higher odds of PPD (COR = 2.92, 95% CI: 1.36 - 6.25, p = 0.01; AOR = 3.58, 95% CI: 1.39 – 9.18, p = 0.01). A history of abortion was also associated with an increased risk of PPD (COR = 2.17, 95% CI: 1.40 - 3.34, p < 0.01; AOR = 1.73, 95% CI: 1.03 - 2.93, p = 0.03). Conversely, unplanned pregnancy appeared to be a protective factor against PPD (COR = 0.65, 95% CI: 0.46 - 0.92, p = 0.01; AOR = 0.66, 95% CI: 0.45 - 0.98, p = 0.04**).** In terms of pregnancy and delivery-related factors, women who experienced complications during pregnancy had significantly higher odds of developing PPD (COR = 2.45, 95% CI: 1.72 - 3.50,

PLOS Mental Health

**Table 2. Reproductive characteristics of participants (N = 540).**

| Variable | Overall N = 540 N (%) | No PPD N = 282 N(%) | PPD N = 258 N(%) |
|---|---|---|---|
| **Parity** | | | |
| Primiparous | 200(37.04) | 120(42.55) | 80(31.01) |
| Multiparous | 340(62.96) | 162(57.45) | 178(68.99) |
| **Age at Menstruation** | | | |
| ≤12 years | 229(42.41) | 117(41.49) | 112(43.41) |
| 13-14 years | 249(46.11) | 130(46.10) | 119(46.12) |
| >14 years | 62(11.48} | 35(12.41) | 27(10.47) |
| **Type of Menstruation before Pregnancy** | | | |
| Regular | 421(77.96) | 231(81.91) | 190(73.64) |
| Sometimes Regular | 85(15.74) | 41(14.54) | 44(17.05) |
| Always Irregular | 34(6.30) | 10(3.55) | 24(9.30) |
| **Age at Marriage** | | | |
| <18 years | 352(65.19) | 184(65.25) | 168(65.12) |
| ≥18 years | 188(34.81) | 98(34.75) | 90(34.88) |
| **Age at First Delivery** | | | |
| <20 years | 334(61.85) | 181(64.18) | 153(59.30) |
| ≥20 years | 206(38.15) | 101(35.82) | 105(40.70) |
| **Number of Children** | | | |
| 1 | 200(37.04) | 120(42.55) | 80(31.01) |
| 2-3 | 316(58.52) | 153(54.26) | 163(63.18) |
| >3 | 24(4.44) | 9(3.19) | 15(5.81) |
| **Birth Interval** | | | |
| ≤24 Months | 115(33.82) | 58(35.80) | 57(32.02) |
| 25-36 Months | 72(21.18) | 31(19.14) | 41(23.03) |
| >36 Months | 153(45.00) | 73(45.06) | 80(44.94) |
| **History of Abortion** | | | |
| Yes | 108(20.00) | 40(14.18) | 68(26.36) |
| No | 432(80.00) | 242(85.82) | 190(73.64) |
| **Type of Abortion** | | | |
| Spontaneous | 65(60.19) | 25(62.50) | 40(58.82) |
| Induced | 16(14.81) | 15(37.50) | 28(41.18) |
| **Methods of Abortion** | | | |
| Spontaneous | 65(60.19) | 25(62.50) | 40(58.82) |
| Medication | 16(14.81) | 7(17.50) | 9(13.24) |
| Menstrual Regulation (MR) | 18(16.67) | 7(17.50) | 11(16.18) |
| Dilatation & Curettage (D&C) | 9(8.33) | 1(2.50) | 8(11.76) |
| **Intention of Last Pregnancy** | | | |
| Planned | 209(38.70) | 123(43.62) | 86(33.33) |
| Unplanned | 331(61.30) | 159(56.38) | 172(66.67) |
| **Pregnancy Term** | | | |
| <36 weeks | 91(16.85) | 46(16.31) | 45(17.44) |
| 36-40 weeks | 389(72.04) | 206(73.05) | 183(70.93) |
| >40 weeks | 60(11.11) | 30(10.64) | 30(11.63) |
| **Twin Pregnancy** | | | |
| Yes | 20(3.70) | 13(4.61) | 7(2.71) |
| No | 520(96.30) | 269(95.39) | 251(97.29) |

*(Continued)*

**Table 2.** (Continued)

| Variable | Overall N = 540 N (%) | No PPD N = 282 N(%) | PPD N = 258 N(%) |
|---|---|---|---|
| **Gender of Neonates** | | | |
| Male | 272(50.37) | 150(53.19) | 122(47.29) |
| Female | 268(49.63) | 132(46.81) | 136(52.71) |
| **Family Support during Pregnancy** | | | |
| Yes | 424(78.52) | 232(82.27) | 192(74.42) |
| No | 116(21.48) | 50(17.73) | 66(25.68) |
| **Perception of Pregnancy** | | | |
| Easy | 125(23.15) | 90(31.91) | 35(13.57) |
| Average | 211(39.07) | 114(40.43) | 97(37.60) |
| Difficult | 204(37.78) | 78(27.66) | 126(48.84) |
| **Perception of Delivery** | | | |
| Easy | 157(29.07) | 106(37.59) | 51(19.77) |
| Average | 135(25.00) | 65(23.05) | 70(27.13) |
| Difficult | 248(45.93) | 111(39.36) | 137(53.10) |
| **Delivery Place** | | | |
| Home | 69(12.78) | 35(12.41) | 34(13.18) |
| Government Facilities | 242(44.81) | 127(45.04) | 115(44.57) |
| Private Facilities | 229(42.41) | 120(42.55) | 109(42.25) |
| **Delivery Methods** | | | |
| Spontaneous Vaginal Delivery | 127(23.52) | 72(25.53) | 55(21.32) |
| Assisted Vaginal Delivery | 65(12.04) | 40(14.18) | 25(9.59) |
| Cesarean Section | 348(64.44) | 170(60.28) | 178(68.99) |
| **Complications During Pregnancy** | | | |
| Yes | 319(59.07) | 138(48.94) | 181(70.16) |
| No | 221(40.93) | 144(51.06) | 77(29.84 |
| **Breastfeeding** | | | |
| Yes | 510(94.44) | 265(93.97) | 245(94.96) |
| No | 30(5.56) | 17(6.03) | 13(5.04) |
| **Transfer of Mother to ICU** | | | |
| Yes | 33(6.11) | 15(5.32) | 18(6.98) |
| No | 507(93.89) | 267(94.68) | 240(93.02) |

p < 0.01; AOR = 2.96, 95% CI: 1.95 – 4.50, p < 0.01). Women who perceived their pregnancy as difficult were over four times more likely to develop PPD than those who had an easy pregnancy (COR = 4.15, 95% CI: 2.65 - 6.72, p < 0.01; AOR = 3.47, 95% CI: 1.85 – 6.49, p < 0.01). Similarly, those who reported a difficult delivery had significantly higher odds of PPD (COR = 2.56, 95% CI: 1.68 - 3.89, p < 0.01; AOR = 2.41, 95% CI: 1.55 - 3.75, p < 0.01). Lack of family support during pregnancy had an increased likelihood of PPD (COR = 0.63, 95% CI: 0.41 - 0.95, p = 0.02; AOR = 0.62, 95% CI: 0.40 - 0.96, p = 0.03**). [**Tables 4–5**]**

### Information of patient attended psychiatric consultation after referral

Of 258 PPD patients, 36.1% attended psychiatric consultations. Attendance was highest at Medical College Hospitals (49.0%), followed by Maternal and Child Health Hospitals (33.3%) and Upazila Health Complexes (23.5%), with a significant difference by facility type (p = 0.01). [Table 6]

PLOS Mental Health

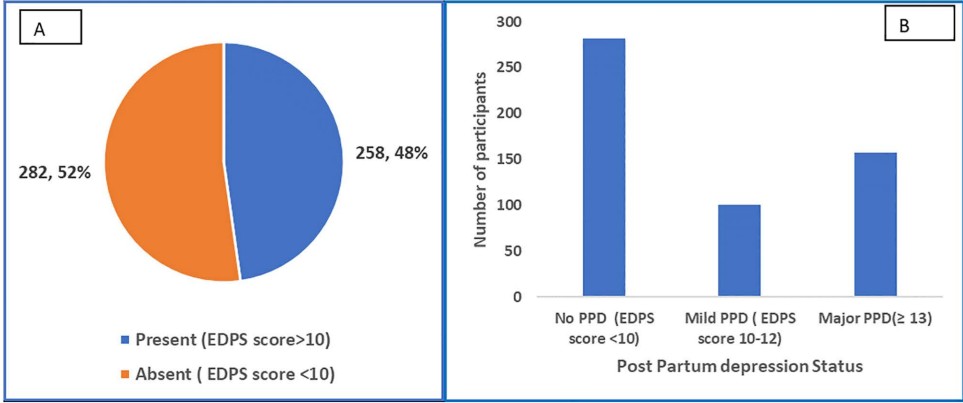

**Fig 1. Prevalence (A) and Severity (B) of Post-Partum Depression (PPD) among Study Participants (N = 540).**

## Discussion

The prevalence of postpartum depression (PPD) in our sample—47.78%, with 29.07% meeting the criteria for major PPD—is notably higher than many earlier reports from Bangladesh. Previous studies conducted in rural regions of the country have estimated PPD rates between 18% and 35%, while a 2019 study conducted among women in Dhaka slums reported a prevalence of 39.4% [26,28,29]. By contrast, one recent urban study in Bangladesh, which included women up to two years postpartum, reported an even higher prevalence of 60.6% [30]. Our higher prevalence rate may reflect several factors, including the timing of postpartum assessment. Unlike some studies that include participants many months or even years after delivery, we restricted recruitment to women between 4 and 12 weeks postpartum. This time frame aligns with the early postpartum period, during which hormonal shifts, sleep deprivation, and the physical and emotional demands of newborn care may heighten vulnerability to depression [31,32].In South Asia, India's overall PPD prevalence is approximately 22% (highest in the southern regions at 26%; 95% CI: 19–32; lowest in the northern regions at 15%; 95% CI: 10–21), and a community survey in Nepal identified PPD in 14.7% of mothers [33,34]. However, among low- and middle-income countries, Afghanistan reported a notably high prevalence of postnatal depression at 60.9% within the first year after childbirth, while a meta-analysis across 59 such countries estimated a pooled prevalence of 24.7% (95% CI: 23.7%–25.6%) [9,35].In contrast, high-income countries typically report prevalence rates ranging between 4% and 13%, with Japan recording the lowest rate at 4% [36,37]. These discrepancies likely reflect differences in inclusion criteria, study designs, screening methods, assessment timing, cultural and socioeconomic contexts, and healthcare access.

The study found that labor-working postpartum women had a higher risk of PPD, possibly due to stress, sleep deprivation, and caregiving burdens. Targeted workplace, clinical, and policy interventions are crucial to support their mental health [38]. Current study found that multiparous women were more likely to experience PPD compared to primiparous, indicating the cumulative emotional and physical toll of multiple pregnancies. This aligns with studies conducted in Japan and Spain, which also reported similar findings [39,40]. Interestingly, some research suggests that grand multiparity is linked to lower PPD screening scores and reduced risk for PPD, with primiparous women exhibiting the highest risk and multiparous women showing intermediate scores [41]. Additionally, we observed that women with a history of abortion were significantly more likely to experience PPD, consistent with other studies highlighting the psychological impact of pregnancy loss on mental health [35]. This can be explained by the fact that a history of miscarriage increases mental stress about the current pregnancy and potential complications [42]. Even minor pregnancy symptoms can trigger anxiety about fetal health, further exacerbating the mother's stress. The findings related to pregnancy intention are noteworthy. Although a majority of participants with unplanned pregnancies reported experiencing PPD, the association was

**Table 3. Univariate regression analysis of sociodemographic characteristics of study participants (N = 540).**

| Variable | No PPD N = 282 N(%) | PPD N = 258 N(%) | COR (95%CI) | p-value |
|---|---|---|---|---|
| **Age Group** | | | | |
| ≤24 | 101(44.29) | 155(54.96) | Reference | |
| 15-29 | 87(30.85) | 88(34.11) | 1.55 (1.05 - 2.29) | 0.02 |
| ≥30 | 40(14.18) | 69(26.74) | 2.65 (1.67 - 4.21) | <0.01 |
| **Religion** | | | | |
| Islam | 272(96.45) | 250(96.90) | Reference | |
| Hinduism | 10(3.55) | 8 (3.10) | 0.87(0.33-2.24) | 0.77 |
| **Educational Level** | | | | |
| No formal education | 16(5.67) | 13(5.04) | Reference | |
| Primary | 35(12.41) | 26(10.08) | 0.91(0.37-2.22) | 0.84 |
| Secondary | 147(52.13) | 138(53.49) | 1.15(0.54-2.49) | 0.71 |
| Higher Secondary | 43(15.25) | 38(14.73) | 1.08(0.46-2.54) | 0.84 |
| Graduate and above | 41(14.54) | 43(16.67) | 1.29(0.55- 3.01) | 0.55 |
| **Occupation** | | | | |
| Housewife | 259(91.86) | 221(85.66) | Reference | |
| Govt. Service | 11(3.90) | 11(4.26) | 1.17(0.49-2.75) | 0.71 |
| Private Service | 6(2.13) | 9(3.49) | 1.75(0.61-5.01) | 0.29 |
| Labor | 6(2.13) | 13(6.20) | 3.12(1.20-8.12) | 0.01 |
| Business | 0(0.00) | 1(0.39) | Omitted | |
| **Husband living Status** | | | | |
| Living Together | 228(80.85) | 203(78.68) | Reference | |
| Dead | 5(1.77) | 4(1.55) | 0.90(0.23-3.39) | 0.87 |
| Living in different place | 16(5.67) | 10(3.88) | 0.70(0.31-1.6) | 0.39 |
| Living Abroad | 33(11.70) | 41(15.89) | 1.39(0.85-2.29) | 0.19 |
| **Family Type** | | | | |
| Nuclear | 152(53.90) | 122(47.29) | Reference | |
| Joint | 130(46.10) | 136(52.71) | 1.30(0.92-1.82) | 0.12 |
| **Husband Educational Level** | | | | |
| No formal education | 24(8.51) | 17(6.59) | Reference | |
| Primary | 38(13.48) | 34(13.18) | 1.26(0.58-2.74) | 0.55 |
| Secondary | 118(41.84) | 111(43.02) | 1.32(0.68-2.60) | 0.40 |
| Higher Secondary | 44(15.60) | 44(17.05) | 1.41(0.68-2.98) | 0.36 |
| Graduate and above | 58(20.57) | 52(20.16) | 1.26(0.61-2.61) | 0.52 |
| **Husband Occupation** | | | | |
| Unemployment | 12(4.26) | 14(5.43) | Reference | |
| Govt. Service | 37(13.12) | 30(11.63) | 0.69(0.28-1.72) | 0.43 |
| Private Service | 78(27.66) | 52(20.16) | 0.57(2.44-1.33) | 0.19 |
| Labor | 57(20.71) | 67(25.97) | 1.00(0.43-2.35) | 0.98 |
| Business | 98(34.75) | 95(36.82) | 0.83(0.36-1.88) | 0.65 |
| **Economic Status (Monthly income)** | | | | |
| Lower class | 130(46.10) | 103(39.92) | Reference | |
| Lower Middle Class | 129(45.74) | 122(47.29) | 1.19(0.83-1.70) | 0.33 |
| Upper Middle Class | 23(8.16) | 33(12.79) | 1.81(1.00-3.27) | 0.04 |
| **Residence** | | | | |
| Rural | 176(62.41) | 166(64.34) | Reference | |
| Urban | 106(37.59) | 92(35.66) | 1.08(0.76-1.54) | 0.46 |

*(Continued)*

**Table 3.** (Continued)

| Variable | No PPD N=282N(%) | PPD N=258 N(%) | COR (95%CI) | p-value |
|---|---|---|---|---|
| **History of Previous Depression** | | | | |
| Absent | 230(81.56) | 147(56.98) | **Reference** | |
| Present | 52(18.44) | 111(43.02) | 3.34(2.26-4.93) | <0.01 |
| **History of Mental Illness in Family** | | | | |
| No | 251(89.01) | 222(86.05) | **Reference** | |
| Yes | 20(7.09) | 27(10.47) | 1.52(0.83-2.79) | 0.17 |
| Don't Know | 11(3.90) | 9(3.49) | 0.92(0.38-2.27) | 0.86 |
| **BMI** | | | | |
| Underweight (< 18.5 kg/m$^2$) | 25(8.87) | 21(8.14) | **Reference** | |
| Normal (< 18.5 kg/m$^2$ to 24.99 kg/m$^2$) | 159(56.38) | 153(59.30) | 1.14(0.61-2.13) | 0.66 |
| Overweight (24.99 to 29.99.kg/m$^2$) | 71(25.18) | 56(21.71) | 0.94(0.48-1.84) | 0.85 |
| Obese (>29.99 kg/m$^2$) | 27(9.57) | 28(10.85) | 1.23(0.56-2.70) | 0.60 |

not statistically significant in our study. This trend is consistent with other research, which found higher PPD prevalence in women with unintended pregnancies compared to those with planned pregnancies (6.7% vs. 4.3%, p<0.01) [43]. Unplanned pregnancies may introduce additional emotional and financial stressors, potentially contributing to the onset of depressive symptoms during the postpartum period. These findings underscore the need to address pregnancy planning as a potential factor in PPD risk.

Our study highlighted that pregnancy complications play a significant role in the development of PPD. Women who experienced difficult pregnancies or deliveries were more likely to develop PPD, likely due to the increased emotional and physical stress linked to these complications. This finding is consistent with other research, which demonstrated that pregnancy complications significantly raise the risk of PPD ($X^2 = 16.45$, df = 1, p<0.001) [44]. These results emphasize the importance of closely monitoring women with complicated pregnancies for signs of PPD and providing appropriate support. Furthermore, our findings regarding family support during pregnancy were consistent with previous research, Women with moderate or low social support were more likely to have postpartum depression with a lack of family support being linked to a higher prevalence of PPD [45]. This highlights the importance of a strong support system during and after pregnancy in mitigating the risk of PPD.

A history of depression prior to pregnancy emerged as a significant predictor of PPD in our study, consistent with previous research findings [46]. This highlights the need for proactive universal screening of mental health conditions during prenatal care to identify women who may be at higher risk of developing PPD. Early identification allows for timely interventions that could potentially reduce the incidence and severity of PPD, emphasizing the importance of integrating mental health assessment tool into routine prenatal care.

Interestingly, while family support appeared to reduce the risk of PPD, this was not statistically significant, indicating that other unmeasured variables may play a role in moderating the relationship between family support and mental health outcomes. Studies conducted on PPD also supported our findings [13,28,40,43,45–47].

In this study, the prevalence of postpartum depression varied across the three tiers of care, which may reflect differences in patient case-mix, referral patterns, and socio-economic characteristics between primary, secondary, and tertiary facilities [22,36].

In this study, women identified as high-risk for PPD (EPDS >13) were counselled, referred to available specialist services, and followed up by phone; however, direct linkage to mental health counsellors was not always feasible due to resource and workforce limitations. Despite the high PPD burden, very few women in our cohort received psychiatric

**Table 4.** Univariate analysis of reproductive characteristics of participants (N = 540).

| Variable | No PPD N = 282 N(%) | PPD N = 258 N(%) | COR (95%CI) | p-value |
|---|---|---|---|---|
| **Parity** | | | | |
| Primiparous | 120(42.55) | 80(31.01) | **Reference** | |
| Multiparous | 162(57.45) | 178(68.99) | 1.65(1.16-2.35) | 0.01 |
| **Age at Menstruation** | | | | |
| ≤12 years | 117(41.49) | 112(43.41) | **Reference** | |
| 13-14 years | 130(46.10) | 119(46.12) | 0.96(0.67-1.37) | 0.81 |
| >14 years | 35(12.41) | 27(10.47) | 0.80(0.46-1.42) | 0.45 |
| **Type of Menstruation before Pregnancy** | | | | |
| Regular | 231(81.91) | 190(73.64) | **Reference** | |
| Sometimes Regular | 41(14.54) | 44(17.05) | 1.30(0.82-2.08) | 0.26 |
| Always Irregular | 10(3.55) | 24(9.30) | 2.92(1.36-6.25) | 0.01 |
| **Age at Marriage** | | | | |
| <18 years | 184(65.25) | 168(65.12) | **Reference** | |
| ≥18 years | 98(34.75) | 90(34.88) | 1.00(0.70-1.43) | 0.97 |
| **Age at First Delivery** | | | | |
| <20 years | 181(64.18) | 153(59.30) | **Reference** | |
| ≥20 years | 101(35.82) | 105(40.70) | 1.23(0.87-1.74) | 0.24 |
| **Number of Children** | | | | |
| 1 | 120(42.55) | 80(31.01) | **Reference** | |
| 2-3 | 153(54.26) | 163(63.18) | 1.60(1.12-2.29) | 0.01 |
| >3 | 9(3.19) | 15(5.81) | 2.50(1.04-5.99) | 0.04 |
| **Birth Interval** | | | | |
| ≤24 Months | 58(35.80) | 57(32.02) | **Reference** | |
| 25-36 Months | 31(19.14) | 41(23.03) | 1.34(0.74-2.43) | 0.32 |
| >36 Months | 73(45.06) | 80(44.94) | 1.12(0.69-1.81) | 0.66 |
| **History of Abortion** | | | | |
| No | 242(85.82) | 190(73.64) | **Reference** | |
| Yes | 40(14.18) | 68(26.36) | 2.17(1.40-3.34) | <0.01 |
| **Type of Abortion** | | | | |
| Spontaneous | 25(62.50) | 40(58.82) | **Reference** | |
| Induced | 15(37.50) | 28(41.18) | 1.17(0.52-2.60) | 0.71 |
| **Methods of Abortion** | | | | |
| Spontaneous | 25(62.50) | 40(58.82) | **Reference** | |
| Medication | 7(17.50) | 9(13.24) | 0.80(0.26-2.43) | 0.70 |
| Menstrual Regulation (MR) | 7(17.50) | 11(16.18) | 0.98(0.34-2.87) | 0.97 |
| Dilatation & Curettage (D&C) | 1(2.50) | 8(11.76) | 5.00(0.59-42.41) | 0.14 |
| **Planned Pregnancy** | | | | |
| No | 159(56.38) | 172(66.67) | **Reference** | |
| Yes | 123(43.62) | 86(33.33) | 0.65(0.46-0.92) | 0.01 |
| **Pregnancy Term** | | | | |
| <36 weeks | 46(16.31) | 45(17.44) | **Reference** | |
| 36-40 weeks | 206(73.05) | 183(70.93) | 0.90(0.57-1.43) | 0.68 |
| >40 weeks | 30(10.64) | 30(11.63) | 1.02(0.53-1.96) | 0.94 |
| **Twin Pregnancy** | | | | |
| No | 269(95.39) | 251(97.29) | | |
| Yes | 13(4.61) | 7(2.71) | 0.93(0.78-1.10) | 0.43 |

*(Continued)*

**Table 4.** (Continued)

| Variable | No PPD N = 282 N(%) | PPD N = 258 N(%) | COR (95%CI) | p-value |
|---|---|---|---|---|
| **Gender of Neonates** | | | | |
| Male | 150(53.19) | 122(47.29) | Reference | |
| Female | 132(46.81) | 136(52.71) | 1.26(0.90-1.78) | 0.17 |
| **Family Support during Pregnancy** | | | | |
| No | 50(17.73) | 66(25.68) | Reference | |
| Yes | 232(82.27) | 192(74.42) | 0.63(0.41-0.95) | 0.02 |
| **Pregnancy time perception** | | | | |
| Easy | 90(31.91) | 35(13.57) | Reference | |
| Average | 114(40.43) | 97(37.60) | 2.18(1.36-3.52) | 0.01 |
| Difficult | 78(27.66) | 126(48.84) | 4.15(2.65-6.72) | <0.01 |
| **Delivery time perception** | | | | |
| Easy | 106(37.59) | 51(19.77) | Reference | |
| Average | 65(23.05) | 70(27.13) | 2.23(1.39-3.60) | 0.01 |
| Difficult | 111(39.36) | 137(53.10) | 2.56(1.68-3.89) | <0.01 |
| **Delivery Place** | | | | |
| Home | 34(13.18) | 35(12.41) | | |
| Government Facilities | 115(44.57) | 127(45.04) | 0.93(0.54-1.59) | 0.80 |
| Private Facilities | 109(42.25) | 120(42.55) | 0.93(0.54-1.60) | 0.81 |
| **Delivery Methods** | | | | |
| Spontaneous Vaginal Delivery | 55(21.32) | 72(25.53) | | |
| Assisted Vaginal Delivery | 25(9.59) | 40(14.18) | 0.81(0.44-1.50) | 0.52 |
| Cesarean Section | 178(68.99) | 170(60.28) | 1.37(0.91-2.06) | 0.13 |
| **Complications During Pregnancy** | | | | |
| No | 77(29.84) | 144(51.06) | Reference | |
| Yes | 181(70.16) | 138(48.94) | 2.45(1.72-3.50) | <0.01 |
| **Breastfeeding** | | | | |
| No | 13(5.04) | 17(6.03) | Reference | |
| Yes | 245(94.96) | 265(93.97) | 1.20(0.57-2.54) | 0.62 |
| **Transfer of Mother to ICU** | | | | |
| No | 240(93.02) | 267(94.68) | Reference | |
| Yes | 18(6.98) | 15(5.32) | 1.34(0.66-2.70) | 0.42 |

follow-up. This gap reflects systemic barriers. Bangladesh like many low-resource countries lacks integrated maternal mental health services, so mothers with PPD often go untreated [48]. Patriarchal norms, stigma, and economic dependence limit women's autonomy and discourage them from seeking mental health care. [49].Strengthening future screening programs by integrating structured referral pathways, including on-site or tele-mental health counselling, may enhance follow-up and continuity of care in low-resource settings like Bangladesh

This study was hospital-based, which may limit generalizability to postpartum women who do not seek facility-based care. Women attending hospitals may differ in socioeconomic status, health-seeking behavior, or access to resources, potentially affecting estimates of PPD prevalence. Women with adverse pregnancy outcomes were excluded because acute bereavement or distress associated with severe neonatal illness could influence EPDS scores independently of postpartum depression, potentially confounding results, and consecutive sampling was used, both of which may further limit generalizability. Future community-based studies including these women are warranted to obtain a more representative assessment of post-partum depression.

**Table 5. Multivariate logistic regression to find out factors associated with PPD among study participants (N = 540).**

| Variable | COR (95%CI) | p-value | AOR(95%CI) | P-value |
|---|---|---|---|---|
| **Age (Years)** | | | | |
| ≥24 | Reference | | Reference | |
| 25-29 | 1.55 (1.05 - 2.29) | 0.02 | 0.86 (0.52 - 1.44) | 0.58 |
| ≥30 | 2.65 (1.67 - 4.21) | <0.01 | 1.72 (0.94 - 3.15) | 0.07 |
| **Economic Status (Monthly income)** | | | | |
| Lower class | Reference | | Reference | |
| Lower Middle Class | 1.19(0.83-1.70) | 0.33 | 1.20(0.78-1.83) | 0.38 |
| Upper Middle Class | 1.81(1.00-3.27) | 0.04 | 1.70(0.85-3.38) | 0.13 |
| **Occupation** | | | | |
| Housewife | | | Reference | |
| Govt. Service | 1.17(0.49-2.75) | 0.71 | 1.28(0.46-3.58) | 0.62 |
| Private Service | 1.75(0.61-5.01) | 0.29 | 0.72(0.22-2.35) | 0.58 |
| Labor | 3.12(1.20-8.12) | 0.01 | 5.17(1.70-15.70) | 0.01 |
| Business | Omitted | | | |
| **Previous history of depression** | | | | |
| Absent | Reference | | Reference | |
| Present | 3.34(2.26-4.93) | <0.01 | 3.38 (2.17 − 5.28) | <0.01 |
| **Parity** | | | | |
| Primiparous | Reference | | Reference | |
| Multiparous | 1.65(1.16-2.35) | 0.01 | 1.20 (0.34 − 4.16) | 0.77 |
| **Type of menstruation** | | | | |
| Regular | Reference | | Reference | |
| Sometimes irregular | 1.30(0.82-2.08) | 0.26 | 1.05 (0.61 − 1.82) | 0.84 |
| Always irregular | 2.92(1.36-6.25) | 0.01 | 3.58 (1.39 − 9.18) | 0.01 |
| **Number of Children** | | | | |
| 1 | Reference | | Reference | |
| 2-3 | 1.60(1.12-2.29) | 0.01 | 1.10 (0.41 - 3.00) | 0.84 |
| >3 | 2.50(1.04-5.99) | 0.04 | Omitted | |
| **History of abortion** | | | | |
| No | Reference | | Reference | |
| Yes | 2.17(1.40-3.34) | <0.01 | 1.73 (1.03 - 2.93) | 0.03 |
| **Planned Pregnancy** | | | | |
| No | Reference | | Reference | |
| Yes | 0.65(0.46-0.92) | 0.01 | 1.14 (0.53 - 2.44) | 0.73 |
| **Complication related to pregnancy** | | | | |
| No | Reference | | Reference | |
| Yes | 2.45(1.72-3.50) | <0.01 | 2.96 (1.95 − 4.50) | <0.01 |
| **Family Support** | | | | |
| No | Reference | | Reference | |
| Yes | 0.63(0.41-0.95) | | 0.64 (0.38 - 1.06) | 0.08 |
| **Pregnancy time perception** | | | | |
| Easy | Reference | | Reference | |
| Average | 2.18(1.36-3.52) | 0.01 | 2.09 (1.16 - 3.76) | 0.01 |
| Difficult | 4.15(2.65-6.72) | <0.01 | 3.47 (1.85 − 6.49) | <0.01 |
| **Delivery time perception** | | | | |
| Easy | Reference | | Reference | |
| Average | 2.23(1.39-3.60) | 0.01 | 2.51 (1.41 - 4.47) | <0.01 |
| Difficult | 2.56(1.68-3.89) | <0.01 | 1.77 (1.04 - 3.02) | 0.0 |

**Table 6. Distribution of PPD Patients Who Attended Psychiatric Consultation After Referral by Facility Type (N = 258).**

| Data Collection Site | Attended n(%) | Not attended n(%) | $X^2$ value | P-value |
|---|---|---|---|---|
| Medical College Hospital | 48 (48.98%) | 50(51.02%) | | |
| Maternal and Child Hospital | 25 (33.33%) | 50 (66.67%) | 13.12 | 0.01 |
| Upazila Health Complex | 20 (23.53%) | 65 (76.47%) | | |

## Conclusion

The high burden of postpartum depression (PPD) identified in this study highlights the critical need for systematic integration of mental health services within routine maternal and postnatal care. Significant risk factors including manual labor, prior depression, irregular menstruation, abortion history, pregnancy complications, and negative perceptions of pregnancy and childbirth underscore the vulnerability of postpartum women. In the absence of a formal integrated referral system, screen-positive women in this study were advised to seek psychiatric consultation; however, only about one-third accessed these services, revealing substantial barriers to care. These findings call for the development of an integrated referral and support system linking PPD screening with accessible, affordable, and culturally sensitive mental health services. Policy action should prioritize embedding mental health within maternal health programs and strengthening referral pathways to ensure timely and equitable care for women affected by PPD. Further research is needed to inform effective implementation strategies and to identify facilitators and barriers to service uptake.

## Supporting information

**S1 Table. Distribution of PPD by Data Collection Site (N = 540).**
(DOCX)

**S2 Table. Distribution of participant by EDPS Categories of Depression Severity among different facilities (N = 540).**
(DOCX)

## Acknowledgments

We sincerely acknowledge the invaluable support provided by the Director and the Department of Gynecology and Obstetrics at Mugda Medical College Hospital, the Maternal and Childhood Health Training Institute, and the dedicated team at the Upazila Health Complex in Singair, Manikganj. We extend our special appreciation to Prof. Dr. Mahmudur Rahman, Country Head of EMPHNET Bangladesh and Gretchen Cowman, Epidemiologist

U.S. Centers for Disease Control and Prevention (CDC)

for their thoughtful review of the study protocol and constructive feedback, as well as to Prof. Dr. Mekhala Sarkar of the National Institute of Mental Health, Bangladesh, for her contribution to the translation of the EPDS tool.

## Author contributions

**Conceptualization:** Md Foyjul Islam, Sirajam Munira.

**Data curation:** Md Foyjul Islam.

**Formal analysis:** Md Foyjul Islam.

**Funding acquisition:** Md Foyjul Islam, Sirajam Munira.

**Investigation:** Md Foyjul Islam, Syeda Tasnuva Maria.

**Methodology:** Md Foyjul Islam, Sirajam Munira.

**Project administration:** Md Foyjul Islam, Quazi Ahmed Zaki.

**Resources:** Md Foyjul Islam.

**Software:** Md Foyjul Islam.

**Supervision:** Md Foyjul Islam, Quazi Ahmed Zaki, Tahmina Shirin.

**Validation:** Md Foyjul Islam.

**Visualization:** Md Foyjul Islam.

**Writing – original draft:** Md Foyjul Islam, Sirajam Munira, Syeda Tasnuva Maria, Quazi Ahmed Zaki, Tahmina Shirin.

**Writing – review & editing:** Md Foyjul Islam, Sirajam Munira, Syeda Tasnuva Maria, Quazi Ahmed Zaki, Tahmina Shirin.

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
