## [Decision Letter · Decision Letter 0]

7 Aug 2025

PMEN-D-25-00285

Strengthening Postpartum Mental Health: Integrating a Mental Health Screening Tool into Routine Postnatal Care

PLOS Mental Health

Dear Dr. Islam,

Thank you for submitting your manuscript to PLOS Mental Health. After careful consideration, we feel that it has merit but does not fully meet PLOS Mental Health’s publication criteria as it currently stands. Therefore, we invite you to submit a revised version of the manuscript that addresses the points raised during the review process.

We look forward to receiving your revised manuscript.

Kind regards,

Lambert Zixin Li, Ph.D.

Academic Editor

PLOS Mental Health

Journal Requirements:

Additional Editor Comments (if provided):

Dear Authors,

Thank you for submitting your manuscript to PLOS Mental Health. After review, we invite you to submit a revised version addressing the comments provided by the reviewers.

While your study shows potential, the reviewers have identified several areas that require clarification or improvement. Please submit a point-by-point response detailing how each comment has been addressed, along with a revised version of the manuscript.

We look forward to receiving your revision.

Sincerely,

Lambert Zixin Li, PhD

PLOS Mental Health

Reviewers' comments:

Reviewer's Responses to Questions

**Comments to the Author**

1. Does this manuscript meet PLOS Mental Health’s publication criteria ? Is the manuscript technically sound, and do the data support the conclusions? The manuscript must describe methodologically and ethically rigorous research with conclusions that are appropriately drawn based on the data presented.

Reviewer #1: Yes

Reviewer #2: Partly

Reviewer #3: Yes

2. Has the statistical analysis been performed appropriately and rigorously?

Reviewer #1: Yes

Reviewer #2: Yes

Reviewer #3: Yes

3. Have the authors made all data underlying the findings in their manuscript fully available (please refer to the Data Availability Statement at the start of the manuscript PDF file)?

Reviewer #1: Yes

Reviewer #2: Yes

Reviewer #3: No

4. Is the manuscript presented in an intelligible fashion and written in standard English?

Reviewer #1: Yes

Reviewer #2: Yes

Reviewer #3: Yes

5. Review Comments to the Author

Reviewer #1: A well-conducted and narrated study utilizing the Edinburgh Postnatal depression scale (EPDS), a 10-item questionnaire used to screen for postnatal depression in women. The authors rightfully argue that integrating the EPDS into routine postnatal care will help enhance early detection of at-risk women in low-income countries, such as Bangladesh, and this early detection is paramount for establishing early mental healthcare intervention. The study's aim and methods were detailed. The authors indicated the percentage of global prevalence of postpartum depression and yet made distinctions of variation that can be found due to geographical location, base, ethnicity or regional development.

The cross-sectional study model used for this research was appropriate to encompass a broad representation of women from various sectors of society, cared for in the different tiers of the healthcare system. A time frame for study recruitment was clarified to disclose that recruitment occurred only after ethical approval was obtained. In addition, a clear indication of the criteria for participant exclusion was stated as well.

The method for determining sample size was indicated, with non-response rates adjustments stated. Details of the questionnaire used for data collection were clearly stated, with intervention exclusions initiated for women identified as high risk for developing postpartum depression to seek mental health, a move emphasized by a follow-up call to ensure compliance with the recommendation.

Finally, the authors excellently disclosed that given such a sensitive subject, data collection practices, statistical analysis, and quality assurance measures were well supervised to minimize bias. This is a well-written article worthy of publication.

However, I have a Question for the authors:

What steps, if any, could have been established during your research study period for any woman identified as high risk for postpartum depression (EPDS>13), who does not comply with the recommendation to seek further mental health counselling? Perhaps going the extra step to connect such subjects with an established mental health counsellor would be a valued first-step intervention to help address this issue of strengthening postpartum mental health services in low-income countries like Bangladesh

Thank you so much for inviting me to review this research study

Reviewer #2: Review of PLos One

Thank you for the opportunity to review this manuscript entitled “Strengthening Postpartum Mental Health: Integrating a Mental Health Screening Tool into Routine Post-Natal Care”. Find below my comments:

Title: There is a disconnection between the title and the rest of the body of the manuscript. While the title suggests an implementation study with expected implementation outcomes such as acceptability, adoption, reach, effectiveness, feasibility, sustenance; this was not the case as the manuscript reported only clinical outcomes. The title should tell us about the entire work at a glance. I suggest a modification. For example: “The Burden of Postpartum Depression and its Socio-demographic and Obstetric Correlates among Parturient in Bangladesh: A Cross-Sectional Design”.

Abstract

• Re-write the study aim in the abstract to reflect the suggested title. For example “to determine the proportion of parturient with PPD and its associated factors in the three tiers of Bangladesh Health Facilities”.

• Insert the 95% CI of the reported prevalence

Background:

• PPD is a different entity from post-partum psychosis. It is not necessarily corrects to say that PPD could turn into post-partum psychosis.

• The research gap(s) were not clearly identified.

• In the penultimate paragraph, making a case for integration of screening of PPD does not arise as the index study has no implementation outcomes.

• Kindly re-write the aim in the last paragraph of this section to tally with your methodology. This is not an implementation study.

Methodology

• Why was women whose babies died or on admission for severe illnesses excluded? Don’t you think that excluding these women has implication for your result?

• Why informed the consideration of 5% non-response rate?

• How was the sample size shared across the three tiers of care or was every eligible participant recruited?

• I doubt if what was described is purposive, it appears more like a convenience or consecutive sampling.

• Kindly describe the process of translation of the EPDS into the local language. Translation and training processes is crucial in reliability of interview. It will add rigor to the methods if this is appropriately described.

• Describe EPDS briefly with emphasis on local psychometric properties.

• If the research assistants administered the questionnaire and training was done to improve inter-rater reliability, kindly insert the statistics for the inter-rater agreements (e.g., Cohen Kappa or Intra-class correlation coefficient) as appropriate.

Results

• Calculate the 95% confidence of the prevalence of PPD reported.

• It would have been good to tell us in addition to what was reported, the overall prevalence of PPD at week 4, 5, 6, …12 among those screened at this points.

• Despite indication an implementation study, no implementation outcome (e.g., acceptability, adoption, reach, effectiveness, feasibility, sustenance) was measured.

Discussion

• Kindly comment on the differences in the reported prevalence across the three tiers of care. What do you think is responsible for the differences? This comment should also extend to the severity rates across settings. It appears from your finding that the prevalence and severity are worse in the highest and lowest levels of care. Discuss the possible explanations?

• Limitations: I consider the exclusion of women with pregnancy-related adversities such neonatal deaths or admission as restrictive, as these population may be more prone to PPD. This should be discussed as it may violate the ethical principle of justice.

• The sampling technique (i.e., purposive) may also limit the generalizability of your findings.

Reviewer #3: I have been through the whole manuscript, thank you for the wonderful opportunity to review such an insightful study.

Introduction:

Line 71-72: Barrier description is solid; I would suggest you to briefly highlight cultural beliefs of stigmatized nature or superstition in Bangladeshi Culture, for example, involvement of traditional healers (kabiraj, ojha etc.), village doctors/pharmacy shopkeepers can also delay accessible mental health care (for your convenience, here's a reference: https://doi.org/10.1007/s10597-021-00790-0)

Data Collection:

Since you've mentioned the expected drop out rate is 5%. But please also mention, exactly how many participants dropped-out of the study/there are missing data.

Good Luck

6. PLOS authors have the option to publish the peer review history of their article (what does this mean? ). If published, this will include your full peer review and any attached files.

**Do you want your identity to be public for this peer review?** For information about this choice, including consent withdrawal, please see our Privacy Policy .

Reviewer #1: **Yes: ** NONYE TOCHI AGHANYA MSc, RN, FNP-C

Reviewer #2: No

Reviewer #3: No

---

## [Editor Report · Decision Letter 1]

7 Sep 2025

The Burden of Postpartum Depression and its Socio-demographic and Obstetric Correlates among Parturient in Bangladesh: A Cross-Sectional Study

PMEN-D-25-00285R1

Dear Dr Islam,

We are pleased to inform you that your manuscript 'The Burden of Postpartum Depression and its Socio-demographic and Obstetric Correlates among Parturient in Bangladesh: A Cross-Sectional Study' has been provisionally accepted for publication in PLOS Mental Health.

Best regards,

Lambert Zixin Li, Ph.D.

Academic Editor

PLOS Mental Health

Thank you for thoroughly addressing all reviewer comments. I have carefully reviewed the revised manuscript, your response letter, and the highlighted changes. I find that the paper is of publishable quality. I would also like to especially thank you for engaging with such a timely and important topic.